# Developing content for a virtual reality scenario that motivates quit attempts in adult smokers: A focus group study with art-based methods

**Tosan Okpako**[1]☉*, **Dimitra Kale**[1]☉, **Olga Perski**[1,2,3]☉, **Jamie Brown**[1,4]☉

**1** Department of Behavioural Science and Health, University College London, London, United Kingdom,
**2** Herbert Wertheim School of Public Health and Human Longevity Science, University of California San Diego, La Jolla, California, United States of America, **3** Faculty of Social Sciences, Tampere University, Tampere, Finland, **4** SPECTRUM Research Consortium, Edinburgh, United Kingdom

☉ These authors contributed equally to this work.
* oritsematosan.okpako.21@ucl.ac.uk

**Data Availability Statement:** The codebook and images underpinning analysis of the current study is available in the Open Science Framework

## Abstract

Virtual reality (VR) could be used to deliver messages to smokers that encourages them to attempt quitting. For a VR smoking cessation intervention to be effective, the target population must find the content engaging, relevant, inoffensive, and compelling. Informed by health behaviour theory and narrative transportation theory, this study used focus groups combined with art-based methods (participant sketches) to inform the development of VR content that will appropriately address smokers' beliefs about quitting smoking. Data were analysed using reflexive thematic analysis. Four in-person focus groups (N = 21) were held between July and August 2023. Just under half the sample were from an ethnic minority (42.8%) and women (42.9%), and the mean age was 33.6 years (standard deviation = 15.9). More than half the sample had a low motivation to quit (61.0%). We developed six themes concerning: the VR content suggested by participants, the rationale behind it, its technological execution and potential widescale implementation. Many participants downplayed the health consequences of smoking, prioritising the immediate rewards of smoking over quitting's long-term benefits. Therefore, participants suggested content set in the future, showing the benefits of cessation or the negative consequences of continued smoking. Family members were recommended as supporting VR characters to increase the contents' emotional salience. Participants also suggested graphic content that would trigger anxiety about smoking, suggesting that fear appeals were welcome. Participants wanted a truly novel intervention- not a leaflet about smoking statistics presented through VR. Participants suggested healthcare locations (e.g., doctors' offices) for implementation, as home ownership of VR headsets is low. Also, this would make the VR appear more legitimate as a health intervention (rather than casual entertainment) and could complement in-person advice. Future research will refine the participant-generated ideas with experts in VR design and smoking cessation.

repository, https://osf.io/v3gdb/ (DOI: 10.17605/OSF.IO/V3GDB).

**Funding:** This study and TO are funded by the Medical Research Council's Doctoral Training Programme. The MRC grant number is R/N013867/1. The funders had no role in study design, data collection and analysis, decision to publish, or preparation of the manuscript.

**Competing interests:** JB has received unrestricted funding to study smoking cessation from Pfizer and J&J who manufacture smoking cessation medications. All authors declare no financial links with the tobacco industry or their representatives.

## Author summary

Virtual reality (VR) involves wearing special glasses that show a digital world on a screen. People using VR often feel like they are really in that digital world. VR could be used to make adults want to quit smoking. When creating a healthcare intervention, getting input from the people who will use it is useful. However, little research has been done on this for health-related VR. We held four focus groups with 21 smokers in July and August 2023. They sketched and gave ideas for VR content. Some ideas showed what could happen in the future if you quit or do not quit smoking, and these were made to feel emotional and personal by adding family members as supporting VR characters in their sketches. Others suggested graphic images to make people scared of smoking. Participants thought that the VR would be most useful if smokers could use it in healthcare settings, such as doctors' offices, because not many people have a headset at home. As part of a larger project, we will refine these ideas with VR designers and healthcare workers in a second study.

## Introduction

Smoking cigarettes increases one's risk of developing serious health conditions such as lung cancer, heart disease and COPD [1]. While most smokers are aware of the health consequences of smoking, in many populations relatively few have concrete intentions to quit in the near future. In England, only 15% have a quit attempt planned for within the next three months [2]. However, most individual-level interventions focus on increasing the success rate of quit attempts. Interventions designed specifically for adults, to prompt quit attempts that would otherwise not happen, remain scarce [3]. With advances in technology, virtual reality (VR) could be used to deliver evidence and theory-based content that encourages more attempts to stop smoking.

VR involves wearing a head-mounted display (headset) that presents images of a virtual environment in such a way that the user feels as if they are "really there" [4]. In a previous pilot study testing VR for encouraging quit attempts among unmotivated smokers, the VR group had a greater increase in willingness to quit compared to the two comparator groups (difference in difference of 3.05 and 2.06 respectively, on the contemplation ladder of smoking cessation, that ranges from 0 to 10, $p < 0.0001$) [5]. In another pre/post study, 23 motivated and unmotivated smokers were shown one of two VR scenarios. At one-month follow-up, 14 had made a quit attempt [6]. This current study involves collecting the viewpoints of smokers through focus groups and art-based methods (participant sketches), to inform the content of a new, person-based, VR smoking cessation scenario aimed at increasing motivation to quit in adults.

Many prominent theories and models exist which seek to explain smoking as a behaviour and the requirements needed for that behaviour to change (i.e. making a quit attempt). The determinants of smoking are a mixture of psychological, biological, and sociocultural factors [7]. Theories of smoking cessation differ in their conceptualisations of the relevant importance of these factors and how they are interlinked. Examples of prominent theories include but are not limited to, "The Health Belief Model", "Social Cognitive Theory" "The Theory of Planned Behaviour" and "The Transtheoretical Model (Stages of Change)" [7–9].However, this study uses COM-B and PRIME theory. The COM-B model of behaviour specifies how capability, opportunity and motivation need to be present for a behaviour to take place. These three factors interact with each other as part of a dynamic system, with motivation forming a core part of the model. Motivation refers to the brain processes that energise and direct behaviour [10].

COM-B can be combined with the PRIME theory of motivation (plans, responses, impulses, motives, evaluations) [11]. According to PRIME theory, plans formed to quit smoking are based on evaluations (reflective thoughts on what is good versus bad) of smoking. Evaluations influence, and are influenced by, motives (wants or needs) to continue smoking or quit. Those motives are then influenced by competing impulses and inhibitions to smoke, based on nicotine cravings and social or environmental cues [11,12]. However, an evaluation of something as "good or bad" is insufficient to influence motives–it must generate an emotional response [12]. Therefore, prompting quit attempts in smokers is about creating a sufficiently strong desire to stop smoking at a given moment in time. Ideally, this would result in on-the-spot commitment to some form of action. For example, accepting a referral to a stop-smoking service and/or obtaining nicotine replacement therapy [13].

The benefit of VR is that it can produce cognitive and emotional responses in users, which are potentially stronger than those elicited by traditional media. A recent study demonstrated that VR induced stronger subjective emotional experiences (excitement, nervousness, hostility, and jitteriness) and physiological arousal in users compared to 2D videos [14]. In the context of smoking, previous VR interventions have resulted in a greater increase in motivation to quit, when compared to cigarette package warnings and traditional videos [5].

VR is quickly emerging as an impactful technology in healthcare. By 2035 it is predicted that VR will affect more than 50% of the NHS workforce in the United Kingdom [15]. However, like other digital health interventions such as telemedicine, VR runs the risk of contributing towards digital inequalities. This is when certain groups cannot benefit from technology due to a lack of equipment, skills, or interest [16]. Only 4% of British households own a VR headset and previous survey data suggests that up to 37% of individuals have little interest in VR [17,18]. If a VR-based smoking cessation intervention is to be effective in the long term and have substantial reach, the general population of smokers (including those from socioeconomically disadvantaged groups) must find it acceptable. Individuals can form opinions about a new digital health intervention, before, during and after they have used it [19]. When learning of a new intervention, individuals consider whether it fits into their value system, will be easy to use and if it is likely to achieve the intended outcomes. The impression they form, which may be negative or positive, influences (and is influenced by) their motivation to change [19]. For VR, both the content of the scenario and its technical features should be acceptable.

This study aimed to inform the development of the narrative content of a VR scenario to encourage smokers to quit. Narratives are a method of presenting information (fictional or non-fictional) in a structured way, with a beginning, middle and end. Narratives pose a question, conflict or quest that must be resolved by the end [20]. They will usually have genre conventions (plot expectations based on the story's genre), themes, characters, and settings [20]. According to narrative transportation theory, high-quality narratives can result in viewers feeling as if they are a part of the story, and result in strong emotional responses. This makes the message of the narrative more impactful and more likely to influence beliefs and behaviours [20,21]. Narratives have previously been used to influence individuals about health behaviours such as quitting smoking [21].

The role of acceptability within public health is reflected in the Medical Research Council's (MRC) guidance for the development and evaluation of complex health interventions and the person-based approach (PBA) [22,23]. The MRC guidance recommends formative development work to understand the views, preferences, and behaviour of the target population [22]. Using the PBA provides a framework to ground the development of a VR intervention in the perspectives of the people who will eventually use it, thereby maximising its acceptability and potential efficacy [23]. The PBA also integrates methods from user-centred design and co-design [24]. Co-design involves recruiting participants as "experts of their experience" and co-

creators in the development of a new intervention, so it includes content that is relevant to them and appropriately targets their most salient beliefs [25,26]. Rather than being passive subjects, participants in this study played a key role in generating ideas for a new VR smoking cessation intervention [19]. To facilitate co-design, this study used focus groups in combination with art-based methods. Art-based methods involve tasks such as photography, painting, or drawing. A focus group approach combined with arts-based methods is appropriate in this context because both methods can encourage collaborative dialogue, creativity, sense-making and critical reflection in participants [27].

### Research questions

This study used focus groups to gain insight into what potential VR content will be acceptable to smokers and appropriately address their evaluative beliefs about smoking and quitting. [12,20].

This study addresses the following research questions:

1. As of 2023, what VR content, if any, would address the beliefs of adult smokers and be acceptable to them?

2. In 2023, what were the various views of adult smokers on VR generally and using VR to encourage a quit attempt?

## Methods

### Study design

To enhance transparency, this study was reported in line with the Consolidated Criteria for Reporting Qualitative Studies (COREQ) 32-item checklist [28]. The protocol was pre-registered on the Open Science Framework https://osf.io/v3gdb/.

This study used focus groups in combination with art-based methods. Focus groups have the unique advantage of allowing participants to hear and respond to diverse viewpoints. If the group dynamic works as intended, a conversational momentum is generated which allows underlying opinions, attitudes, and descriptions of past experiences to emerge [29]. The process of elaborating on and justifying views can trigger novel questions and comments by participants to each other, that a researcher in a one-to-one interview may not have thought of [30,31]. Here, the aim of the focus groups was not to force consensus among participants but to elicit a range of experiences and ideas [30]. However, the limitation of focus groups is that participants may censor themselves around a group if a sensitive topic is discussed or if they have a minority opinion [32].

Art-based methods such as drawing have previously been used in combination with traditional qualitative data collection methods [33]. The benefit of art-based methods is that they encourage participants to think "outside the box" [34]. They can help uncover tacit knowledge and subliminal thoughts not easily expressed through words [35]. Art-based methods can also make focus groups more engaging and enable less verbose participants to still contribute. A drawback is that participants might become nervous about their artistic abilities being judged [35].

### Qualitative framework

We used a critical-realist theoretical framework to inform data collection and analysis, selected as it can be applied to qualitative research to provide causal explanations, which is useful for intervention development [36]. It makes explicit the assumptions, which are below the surface, that are embodied in everyday activities and underpin practices. These assumptions are often

unreflected and tacit. Multiple assumptions can be held simultaneously and may contradict each other. [37]. A critical-realist approach prevents oversimplification and promotes a deeper understanding of complex social phenomena.

## Participant selection

**Method of approach.** Participants were recruited through digital posters advertised on Facebook and Instagram, Callforparticipants.com and physical posters displayed around University College London's (UCL) campus. The poster included a link to an online screening survey (S1 Table), which was administered via the REDCap electronic data system (embedded within UCL's Data Safe Haven) [38].

**Inclusion and exclusion criteria.** To be eligible for the focus groups, participants had to currently smoke, speak English, be ≥18 years old and able to travel to UCL. People were eligible to participate regardless of their motivation to quit smoking. Some individuals make quit attempts with no prior planning, and these have a similar probability of success to pre-planned quit attempts [13]. Behaviour and motivation are dynamic processes and survey data have shown that up to 13% of individuals who report no intention to quit smoking go on to make a quit attempt in the next 6 months [39]. PRIME theory promotes offering cessation support to smokers, regardless of their current level of motivation to quit [13]. This is in contrast to the transtheoretical model, which posits that smokers with different readiness to change (stages) require different interventions. For example, only those who are most ready to quit should receive advice on cessation medication [8,40]. Therefore, the views of all smokers were relevant to this study. All eligible participants, who completed the survey, were contacted by email to schedule a group slot. They were also contacted by phone a day before the focus group to confirm their attendance.

**Sample size.** Recommendations for focus group size suggest six to eight participants per session [30]. This study aimed to recruit six participants for four different focus groups (N = 24), to achieve theoretical saturation (i.e. when the collection and analysis of additional data does not provide new insights or themes). However, the small sample size (which is common in qualitative research) means that the data obtained may not fully represent the views of the entire target population [31]. Therefore, this study sought a maximum variation sample and used a purposive sampling strategy. The goal was to include a range of different ages, genders, levels of motivation to quit, daily cigarette consumption and different past experiences of VR, with at least half the sample coming from a disadvantaged socioeconomic position (SEP). The survey collected data on these demographic and smoking characteristics.

**Measures.** Motivation to quit was measured using the motivation to stop scale (MTSS), which was validated in the Smoking Toolkit Study- a national, monthly cross-sectional survey in England [39]. High motivation was defined as responses of "I really want to stop and intend to in the next one month" and "I really want to stop and intend to in the next three months" (responses 6–7) [41]. All other responses (1–5) were categorised as low motivation to quit. In a previous study to assess the MTSS' predictive validity among the English smoking population, the accuracy to discriminate between smokers who did and did not attempt to quit had an area under the curve ($ROC_{AUC}$) of 0.67 (95% CI = 0.65–0.70), which is considered to be broadly acceptable [39]. Research comparing The Transtheoretical Model (Stages of Change), the MTSS, and a Likert scale showed comparable predictive and construct validity as measures for intention to quit smoking [42].

Education and occupation were used as proxy measurements for SEP. For this study, a disadvantaged SEP was defined as not having post-16 educational qualifications (e.g. A-levels or a university degree) and/or having a manual occupation. Education captures the long-term

influence of early life circumstances (childhood resources and opportunities) and is correlated with income [43]. Additionally, lower educational attainment is associated with reduced digital literacy [16]. Occupation is also associated with income and education and may reflect access to material resources such as owning advanced technology like a VR headset [43].

## Setting

The focus groups were conducted in person at the University College London campus in London, between July and August 2023. T.O. facilitated the group discussions and an MSc student (T.C.) monitored the recording equipment and took observational notes.

## Data collection

A question route, using the structure recommended by Kreuger and Casey [44], steered the discussions. Before the focus groups, a patient and public involvement (PPI) group checked the questions for clarity and relevancy, and we made modifications to the question route accordingly (S2 Table). It began with more general, fact-based questions and transitioned into experienced-based and open-ended questions designed to generate informal conversation. Open-ended questions give participants flexibility in answering and can generate novel responses [45]. The question route was formatted to unfold over five main phases:

- *Phase 1- Introductions*: The facilitator (T.O.) gave a briefing to establish guidelines for conversation and confidentiality, and to reiterate the study aims.

- *Phase 2-VR demonstration*: As some participants may not have used VR before, we wanted to briefly familiarise them with the technology. Participants played one of two VR games. The first was a 3D painting game in an abstract virtual environment called "Multibrush," on PICO 4 Enterprise [46] The PICO 4 Enterprise headset retails at approximately £900 [47]. The second was an interactive video about marine wildlife called "Hydrous presents: EXPLORE" on HTC Vive [48] This virtual environment had realistic images of the ocean. The HTC Vive headset costs £1,299 [49] These headsets were previously owned by the research team. Both contain the standard features of VR such as a screen with a 360-degree view and hand controllers that allow the user to interact with the virtual environment. However, both VR games are available on cheaper VR headsets, such as Oculus Meta Quest 2, which costs £250 on average [50] Each participant chose which demonstration they would prefer and spent approximately 5–7 minutes playing their chosen game. Participants were asked about their reactions to the VR demonstration, including what they did and did not like and if it was easy or difficult to use.

- *Phase 3- Beliefs about quitting smoking*: In this part of the discussion, the facilitator asked questions based on PRIME theory. PRIME theory has previously been used in focus groups to examine motives to continue or stop smoking in adults who smoke [51, 52]. The questions focused on the participants' past experiences of quitting smoking, including current and past motivators and barriers to quitting.

- *Phase 4- Co-design*: The facilitator provided participants with markers and paper and asked them to use the discussion points raised in phases 2 and 3 to draw ideas for VR still frames they thought would increase their motivation to quit smoking. Participants could write bullet points if unable to draw. Participants shared their designs verbally with the group. Per narrative transportation theory, this phase looked at what types of characters, themes and settings were seen as interesting, exciting, and inoffensive.

- *Phase 5- Debriefing*: The facilitator clarified any inaccurate information stated in the course of the discussions (for example correcting misconceptions around alternative nicotine therapies) (18). The debriefing allowed the participants to ask any questions related to study procedures. Participants were emailed a £20 gift card as compensation for their time and were given refreshments during the focus group sessions.

Each focus group session lasted one and a half to two hours. The sessions were audio-recorded and transcribed by T.O. Observational notes were taken during the sessions to capture body language and non-verbal communication and these notes were added to the transcript. We added the digitally scanned sketches to the transcripts. The transcripts were not returned to participants for comment. However, participants were emailed a plain-language summary of the results.

## Analysis

The data was analysed using reflexive thematic analysis underpinned by critical realism. Analysis was conducted in six phases, as outlined by Clarke and Braun which include: 1. Familiarisation with the data, 2. Generating initial codes, 3. Searching for themes, 4. Reviewing themes, 5. Defining and naming themes and 6. Producing the report [53,54]. Data was coded by T.O., who used a mixture of deductive codes (based on PRIME and narrative transportation theory) and inductive (data-driven) codes. A hybrid approach that combines deductive and inductive coding can add rigour to qualitative analysis as it allows the use of theory while still rooting the results in the raw data [55]. Also, in line with critical realism, the use of inductive and deductive codes aids in abduction- the process of redescribing what is observed (participant responses) in terms of theory to provide plausible causal explanations [56].

We also used semantic (literal and surface readings of data) and latent codes (looking at the underlying ideas shaping responses). Each code was labelled and defined, with example quotes from the transcripts. When coding the sketches, we looked at the main and supporting characters (semantic and inductive) and the genre conventions (latent and deductive). An example of a coded sketch is in the supplementary material (S1 Fig). The initial codebook was discussed with the wider research team (D.K., O.P., and J.B.) and then subsequently refined.

The codes were grouped into themes and the themes were also discussed with the wider research team. We checked each theme for dependability (the consistency with which interpretations of the same data are made by different researchers) and confirmability (the extent to which the findings are the product of the data and not the researcher's biases) of the results. These conversations also promoted reflexivity in the respective researchers [54,57]. To minimise bias in the interpretation of results, the wider research team, highlighted which codes needed more evidence from quotes in the transcripts. NVivo version 12 was used to assist in data management and reorganisation [58].

## Reflexivity

**Personal characteristics of interviewers.**   This study forms a part of T.O.'s PhD thesis. T. O. does not smoke and is a female PhD student with an MSc in Public Health and previous experience conducting interviews. T.O. facilitated the focus groups and was the primary analyst of the transcripts. For this study, T.O. underwent additional training in interviewing and thematic analysis.

**Relationship with participants and to topic.**   The identities and characteristics of the researcher/ moderator and the research participants influence the data generation process [59]. However, multiple identities can be held at the same time. Study participants did not

know T.O. or the wider research team and there was little contact before the focus groups except for a reminder email or call. Regarding T.O.'s positionality within the newly established researcher-participant relationship, the sample was predominantly younger and a similar age to T.O. which may have helped build rapport faster within the groups and reduce the appearance of any power imbalances. However, some participants (who directly asked) knew that T.O. was a non-smoker. At times, this may have placed her as an "outsider" within that social dynamic.

Participants had some knowledge of the study so they could provide informed consent to participate. However, during the focus groups, T.O. tried to de-emphasise the project's overall goal (developing a VR smoking cessation intervention) and emphasised that both negative and positive answers were welcome. This was to prevent participants from feeling obligated to give favourable answers (social desirability bias). Occasionally, the moderator gently asked quieter participants if they would like to contribute, but otherwise left the conversations to proceed organically. Participants were also given ground rules before the discussion, regarding respecting alternative viewpoints and listening to others. They were also reassured that if they became distressed, they could leave the conversation at any time and still receive compensation for their time. No participants left the focus group prematurely and no adverse events were reported.

### Changes to pre-registration

In the pre-registered protocol, we planned for the focus groups to be segmented by motivation to quit (high versus low). Due to recruitment and scheduling constraints, we were unable to do this.

### Ethical approval

Ethical approval was granted by UCL's Research Ethics Committee (ID: 25627.002). Participants provided informed consent. Identifying information, such as participants' names, was removed and data was stored securely. Participants had the option to collect smoking cessation leaflets and business cards after each focus group. Participants were given the contact details of the UCL ethics committee to report any concerns. No concerns or complaints were reported after the focus groups.

## Results

### Participants

Seventy individuals completed the screening survey and 57 were eligible and provided valid contact details and were emailed to schedule a group slot. The final sample had 21 participants, with individual focus groups ranging from 2–10 participants due to cancellations on the day and oversampling (Fig 1).

Table 1 shows the smoking, demographic, and digital experience characteristics of the sample. There was a largely even gender split (42.9% women). The sample was predominantly young, with a mean age of 33.6 years (standard deviation = 15.9). The sample was ethnically diverse, with just under half coming from an ethnic minority. The majority of the sample was not highly motivated to quit smoking (61.0%). Regarding SEP, we classified 38.1% of the sample as having a less advantaged socioeconomic position (either a manual job or no post-16 educational qualifications). Just under half the sample (42.9%) had used VR more than once. The smaller focus groups (groups 1 and 4) were more homogenous than the larger ones in terms of gender, age, and educational attainment.

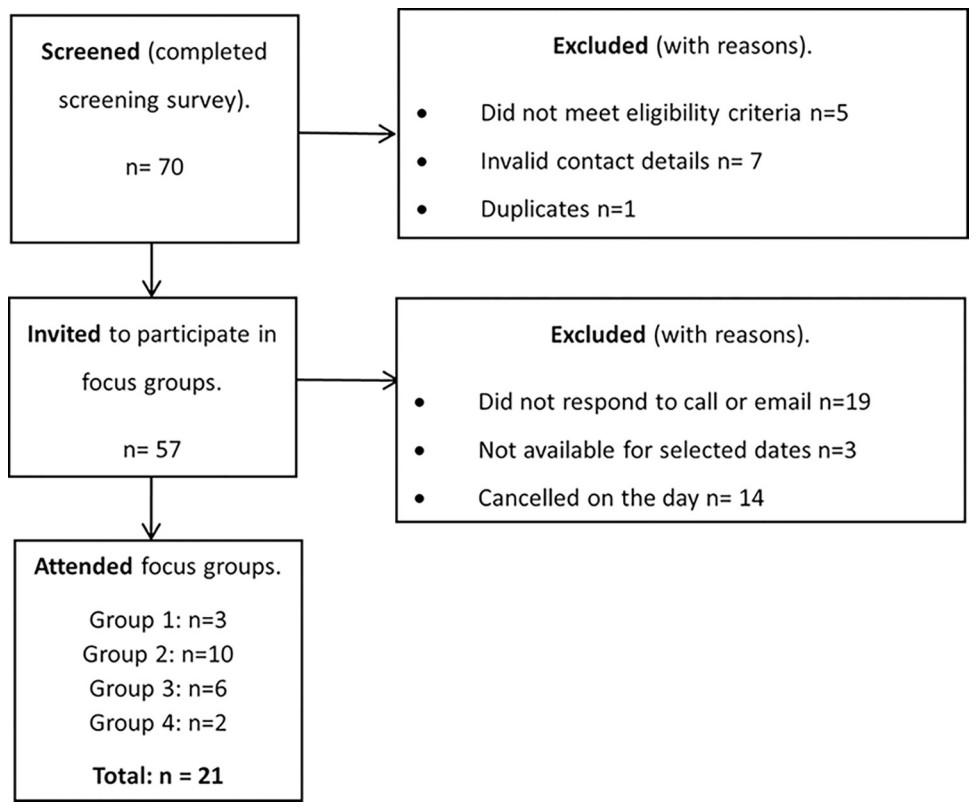

**Fig 1. Participant selection diagram.**

## Themes

**Overview.** We created a total of 35 codes, which we used to develop six themes. Table 2 shows how each code sits within each theme. A full description of each code, and all the quotes and images supporting that code, can be found at https://osf.io/v3gdb/ [60]

The theme related to the VR content rationale "*The rewards from smoking today outweigh the potential health consequences of the future*" is about how most participants did not take seriously the negative health effects of smoking. The causal assumption is that this informed the content suggested by the participants. There were two main themes about the VR content participants wanted to see in a VR scenario aimed at prompting quit attempts. These were, "*Using personalised "coming of age" stories to elicit hope or regret*" and "*Using horror stories to elicit fear*". Therefore, the VR scenario must either bring about an emotional response connected to family and/ or growing older or generate a large amount of fear and anxiety. The theme related to VR content execution "*Exploit VR's technological capabilities*" explains how both types of content should make use of the novel technological qualities of VR.

Regardless of the content suggested, most participants were amenable to the idea of using VR to prompt cessation and envisioned it as a type of health service treatment offered in primary care settings (theme- "*Smoking cessation VR as a health service treatment*"). However, some participants had concerns about how the VR would fit into long-term cessation support. A quit attempt once prompted, needs to be sustained. Participants discussed struggles they had with previous quit attempts and the type of advice they would like to receive. While beyond the scope of this specific project, some participants suggested long-term VR programs

**Table 1. Sample characteristics.**

| Demographic and smoking characteristics | Overall N = 21, | Group 1 n = 3 | Group 2 n = 10 | Group 3 n = 6 | Group 4 n = 2 |
|---|---|---|---|---|---|
| **Gender n (%)** | | | | | |
| Men | 10 (47.6) | 3 (100.0) | 5 (50.0) | 2 (33.3) | 0 (0.0) |
| Women | 9 (42.9) | 0 (0.0) | 5 (50.0) | 2 (33.3) | 2 (100.0) |
| In another way/ prefer not to say | 2 (9.5) | 0 (0.0) | 0 (0.0) | 2 (33.3) | 0 (0.0) |
| *Age (years)* **n** *(%)* | | | | | |
| Mean (SD) | 33.6 (15.9) | 43.7 (2.5) | 33.6 (20.8) | 27 (8.5) | 38 (1.4) |
| 18–24 | 9 (42.9) | 0 (0.0) | 6 (60.0) | 3 (50.0) | 0 (0.0) |
| 25–34 | 2 (9.5) | 0 (0.0) | 1 (10.0) | 1 (16.7) | 0 (0.0) |
| 35–44 | 7 (33.3) | 2 (66.7) | 1 (10.0) | 2 (33.3) | 2 (100.0) |
| 45–54 | 2 (9.5) | 1 (33.3) | 1 (10.0) | 0 (0.0) | 0 (0.0) |
| 55–64 | 0 (0.0) | 0 (0.0) | 0 (0.0) | 0 (0.0) | 0 (0.0) |
| 64+ | 1 (4.8) | 0 (0.0) | 1 (10.0) | 0 (0.0) | 0 (0.0) |
| *Ethnicity n (%)* | | | | | |
| Any Asian or Asian British background | 6 (28.6) | 1 (33.3) | 3 (30.0) | 2 (33.3) | 0 (0.0) |
| Any Black, Black British, Caribbean, or African background | 2 (9.5) | 1 (33.3) | 0 (0.0) | 0 (0.0) | 1 (50.0) |
| Any white background | 12 (57.1) | 1 (33.3) | 6 (60.0) | 4 (66.7) | 1 (50.0) |
| Other ethnic group (e.g. Arab) | 1 (4.8) | 0 (0.0) | 1 (10.0) | 0 (0.0) | 0 (0.0) |
| *Occupational status n (%)* | | | | | |
| Manual | 7 (33.3) | 2 (66.7) | 2 (20.0) | 3 (50.0) | 0 (0.0) |
| Non-manual | 5 (23.8) | 1 (33.3) | 2 (20.0) | 1 (16.7) | 1 (50.0) |
| Student | 6 (28.6) | 0 (0.0) | 5 (50.0) | 1 (16.7) | 0 (0.0) |
| Other (retired, unemployed etc.) | 3 (14.3) | 0 (0.0) | 1 (10.0) | 1 (16.7) | 1 (50.0) |
| *Educational attainment n (%)* | | | | | |
| Post-16 qualifications | 18 (85.7) | 3 (100.0) | 9 (90.0) | 4 (66.7) | 2 (50.0) |
| No post-16 qualifications | 3 (14.3) | 0 (0.0) | 1 (10.0) | 2 (33.3) | 0 (0.0) |
| *Daily cigarettes consumption n (%)* | | | | | |
| 10> | 14 (66.7) | 2 (66.7) | 7 (70.0) | 4 (66.7> | 1 (50.0) |
| 11–20 | 6 (28.6) | 0 (0.0) | 3 (30.0) | 2 (33.3) | 1 (50.0) |
| 21< | 1 (4.8) | 1 (33.3) | 0 (0.0) | 0 (0.0) | 0 (0.0) |
| *Motivation to quit smoking (derived from MTSS) n (%)* | | | | | |
| High | 8 (38.1) | 3 (100.0) | 2 (20.0) | 2 (33.3) | 1 (50.0) |
| Low | 13 (61.9) | 0 (0.0) | 8 (80.0) | 4 (66.7) | 1 (50.0) |
| *Previously used a digital smoking cessation aid n (%)* | | | | | |
| Yes | 9 (42.9) | 2 (66.7) | 5 (50.0) | 1 (16.7) | 1 (50.0) |
| No | 12 (57.1) | 1 (33.3) | 5 (50.0) | 5 (83.3) | 1 (50.0) |
| *Previously used VR more than once n (%)* | | | | | |
| Yes | 9 (42.9) | 2 (66.7) | 4 (40.0) | 1 (16.7) | 2 (50.0) |
| No | 12 (57.1) | 1 (33.3) | 6 (60.0) | 5 (83.3) | 0 (0.0) |

focused on educational and behavioural support (theme: "*What happens after: Using VR for long-term cessation support*").

**Theme 1- VR content rationale: The rewards from smoking today outweigh the potential health consequences of the future**

The underpinning rationale for suggestions on the VR content related to how immediate rewards from smoking outweigh future health consequences. For example, most participants were aware of the negative health consequences of smoking, and some had negative feelings

**Table 2. Themes and the related codes concerning the study's main research questions.**

| Theme | Related codes | |
|---|---|---|
| **Research Question 1: As of 2023, what VR content, if any, would address the beliefs of adult smokers and be acceptable to them?** | | |
| Theme 1-VR content rationale: The rewards from smoking today outweigh the potential health consequences of the future. | • Theoretical consequences<br>• Perpetually postponed or no plans to quit.<br>• Positive evaluative judgements | • Negative/ positive smoker identity<br>• Always trying to quit<br>• Real and virtual health consequences. |
| Theme 2- VR content: Using personalised "coming of age" stories to elicit hope or regret. | • Content that looks into the future<br>• Content- benefits of cessation<br>• Yourself as the VR protagonist<br>• Clinical virtual environment<br>• Family influences on smoking | • Family as supporting VR characters.<br>• Real and virtual health consequences<br>• Tailoring for each VR user |
| Theme 3- VR content: Using horror stories to elicit fear. | • Nightmarish content<br>• Trigger warnings | • Increasing VR immersion<br>• Real and virtual health consequences |
| Theme 4- VR content execution: Exploit VR's technological capabilities. | • Tailoring for each VR user<br>• Increasing VR immersion | • Not "gimmicky"<br>• Not boring cessation leaflets and advertisements |
| **Research Question 2: In 2023, what were the various views of adult smokers on VR generally and using VR to encourage a quit attempt?** | | |
| Theme 5- Smoking cessation VR as a health service treatment | • Clinical/ community implementation locations<br>• Evidenced-based VR<br>• Safeguarding<br>• VR enjoyment | • VR as an adjunct<br>• Dissatisfying encounters with cessation professionals and services<br>• Barriers to using VR at home<br>• Renting VR headsets |
| Theme 6- What happens after: Using VR for long-term cessation support. | • Social life and smoking<br>• Environmental and mental cues<br>• Motives- wants and needs<br>• Educational content<br>• Smoking alternatives | • VR is a temporary fix<br>• Habits<br>• Always trying to quit<br>• Preferred advice- cutting down<br>• Long-term VR meditational content |

towards themselves as someone who smokes, which sometimes included feelings of shame. Only one participant reported having a positive smoker identity.

*"I was actually ashamed of being a smoker"*- **P87, 25–34, low motivation, woman**.

*"Like, whatever intervention you get, nothing replaces that [smoking] and it becomes part of your personality in a way. It's almost like, you can't imagine yourself living your life without smoking."*- **P103, 18–24, low motivation, man**.

*"The way I feel about smoking depends on what the weather is. I don't vape so I can't do it inside. When it's cold and you can't even light the cigarette, you feel like such a loser"* -**P168, 18–24, low motivation, man.**

In contrast to

*"I was like fourteen and I just chugged the whole thing without coughing once and it felt really smooth, and I thought, "This is my niche, this is what I'm good at"."*- **P109, 18–24, low motivation, man.**

Despite knowing the health effects of smoking (e.g., smoking increases the risk of lung cancer), for many participants these health consequences felt theoretical, "far away", and part of a distant, abstract future and did not need to be taken seriously at the present moment.

*"But when the threat is so far into the future, like you're told how you're more likely to develop lung cancer, at this age I'm not going to think about whatever health problems I'll be facing*

*when I'm older. So, it doesn't seem like a "now" problem"* -**P136, 18–24, low motivation, woman**.

*"The only time I thought about stopping completely was thinking that it won't be good for my health, so trying to force myself to do that, but then I decided "What the heck, I'll just do it anyway.""*- **P157, 35–44, high motivation, man.**

This feeling was particularly strong among the younger participants (<44 years), who still felt themselves to be in good health. De-emphasising the health effects meant that many participants were always making plans to quit, but never starting them- they were perpetually postponed.

*"There's always "I'll wait till the end of school, or I'll wait until after results or after the holidays or when I begin class again," and it never comes. That point will never come, I'll just keep pushing it. There's always "When I finish this pack I'll quit," but then I buy another pack. Then I say, "When I finish this cigarette pack I'll quit", but then it becomes a cycle."*- **P163, 18–24, high motivation, other gender identity**.

*"Yeah, honestly no one says, "I'm going to smoke until I die". People do say, in the back of their head, that they want to stop. . . For the past two years, I've been telling myself I'm going to stop."* -**P3, 35–44, high motivation, man.**

However, across most age groups, participants who had experienced real-life health scares or had a family member who did, treated the health effects of smoking seriously.

*"I'm having treatment for [redacted- dental condition] and thirty years of smoking has damaged my gums. My gums are weak, and I think "My goodness- if I carry on, then this is just the start. What else next?" So, it's been a big shock to me . . ..it really freaked me out".* -**P19, 45–54, high motivation, woman.**

As this theme explains, the rationale behind themes 2 and 3 on the VR content was that it could be used to fast-track this moment of "awakening" (Fig 2).

**Theme 2- VR Content: Using personalised "coming of age" stories to elicit hope or regret.** The VR content suggested by participants went beyond the educational goal of simply relaying statistics around cancer and lung disease, to proposing more experiential content that felt personal to the user. Fig 3 shows a selection of sketches drawn by participants (the full collection is available here: https://osf.io/v3gdb/ DOI: 10.17605/OSF.IO/V3GDB). We interpreted these sketches as being influenced by the genre conventions of "coming of age" stories. Traditional "coming of age" narratives follow a protagonist from childhood into adolescence and then adulthood. Here, the participants drew themselves as the main character, often in scenarios set further into the future, representing a delayed "coming of age" (from early adulthood into late adulthood). In these future scenarios, the participant had either drawn a positive situation in which they had quit smoking (Fig 3, sketches B and C), a negative situation in which they had not quit smoking (Fig 4, sketch D) or both (Fig 3, sketch A).

The goal of the positive scenarios was to encourage the user and generate feelings of hope. Contrastingly, the goal of the negative scenarios was to generate feelings of regret about life events they missed and/ or the negative effects of smoking on their future health and appearance. There was variation among participants regarding whether a positive or negative scenario would be more effective. For example:

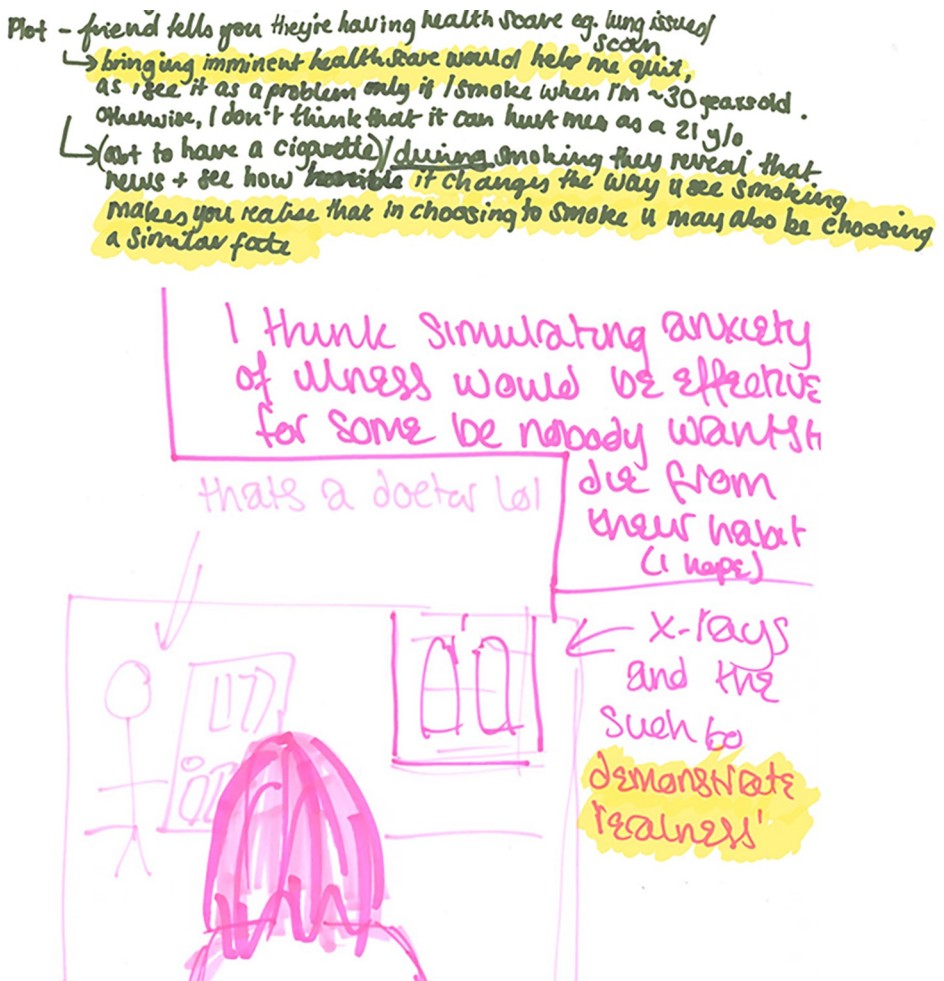

**Fig 2. Written rationales behind suggested VR content.** Top is participant 137, 18–24, low motivation. Bottom is participant 168, 18–24, high motivation. Highlights added.

*"Hmmm, also they [current cessation approaches] don't often focus on, well slightly, but they need to address the benefits. Like, you don't often see the benefits of quitting that happen over time. You don't really think of that. Like what happens when you quit for one year, two years, five years. . . It just sounds good, in contrast to the stuff that's just telling you how bad it is. We already know it's bad."* -**P162, 18–24, low motivation, man.**

*[Describing their sketch] "Also, as an incentive you can see your "lung bar" increasing. That's probably not an incentive enough for most people, but there has to be some sort of incentive that you can see in front of you, otherwise you have the "stick" but no "carrot""*-**P168, 18–24, low motivation, man.**

In contrast to:

*"If it was something drastic, showing you getting really ill then maybe I'd be interested, maybe that would make an impression, but otherwise I doubt it would work."* -**P119, 25–34, low motivation, woman.**

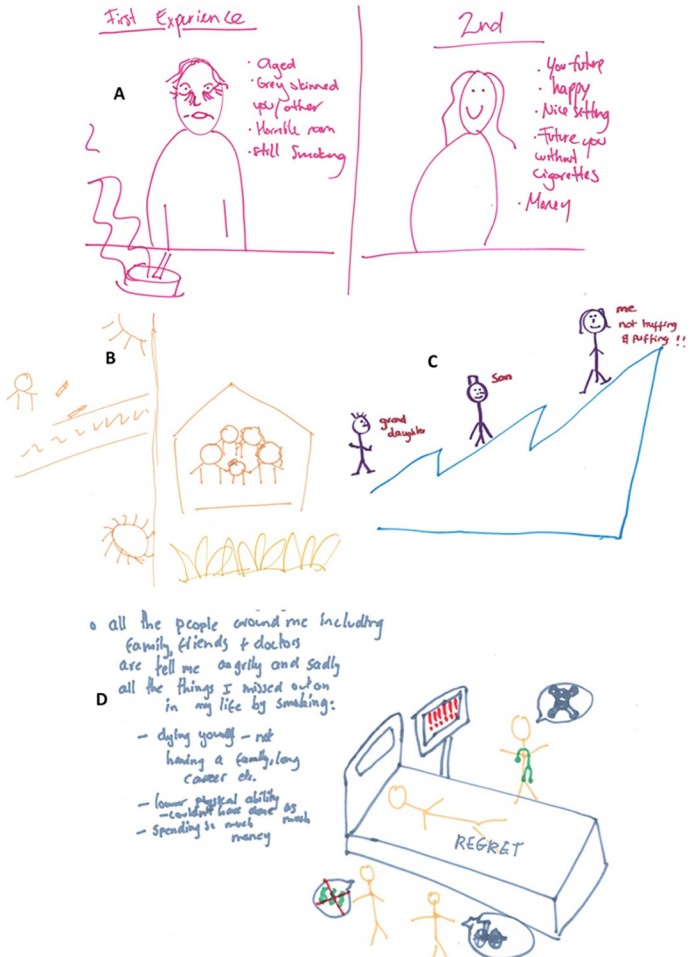

**Fig 3. Theme 2- Personalised "coming of age" stories image selections.**

Furthermore, in line with "coming of age" stories, in much of the suggested content, family members were recommended as supporting characters (Fig 3, sketches, B, C and D). This likely reflects how for the majority of participants, their motivation to quit had changed over time, due to changing family circumstances. For example, participants who had become parents expressed that their motivation to quit had increased because they wanted to live longer for their children and grandchildren.

*"Yeah, for me, the main difference is that I'm a family man now and obviously with a daughter, it's centred around her."* -**P17, 35–44, low motivation, man.**

Seeing younger family members start smoking was also a factor that changed a small number of participants' views towards smoking.

*"I have little brothers and they both started to smoke. . . It's quite shocking when you have somebody who you care about smoking and your different attitude towards that. . . I see them smoking now and I'm like, "Why did I do that?""* -**P165, 35–44, high motivation, woman.**

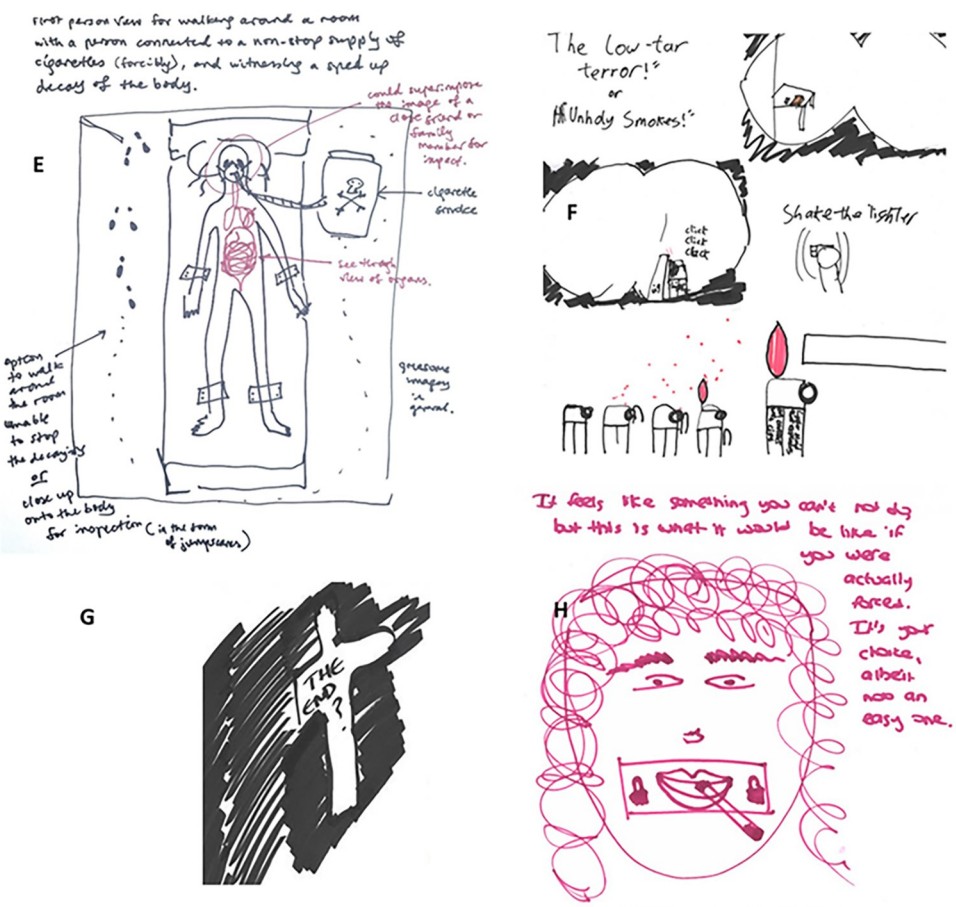

**Fig 4. Theme 3- Horror stories image selections.**

Some participants felt it would be "disrespectful" to continue smoking when family members had become ill. Also, some participants felt pressure from family members (such as partners and parents) to quit.

*"I heard from my mum that my dad has some health problems. Since I can't do anything here, I don't want to be destroying my lungs, while my dad is having health problems". -***P92, 18–24, low motivation, woman.**

However, within the focus groups, a small minority of participants openly disagreed with this line of thinking and felt that the primary motivation for quitting should be for themselves.

*"I do feel like you have to want to quit. It's not going to be enough that a relative has cancer. It has to really come from you." -***P108, 35–44, low motivation, woman**.

*"With my most successful attempts, the motivation has always been wanting to feel better. For me, it was never for anyone else. I would fail if the reason was someone else expressing interest in me quitting." -***P168, 18–24, low motivation, man.**

**Theme 3- VR Content: Using horror stories to elicit fear.** This theme also describes a popular type of content suggested by participants. The primary focus was to use the VR scenario to elicit fear and anxiety. In contrast to the previous theme where the negative scenarios

were grounded in events that could realistically take place (for example, getting cancer), here, the content took a surrealist and exaggerated turn. Fig 4 shows a collection of sketches that illustrate this theme. Examples of content drawn include being in a room where cigarette ash falls on you and having cigarettes padlocked onto your mouth (Fig 4, sketch H). Other descriptions include:

*"So, you walk in, see the cigarettes and stuff on the table and you shake the lighter and there are all these effects, and, in the end, it blows off your head. . . It's kind of like to scare you or traumatise you. . .I think if you saw the flame chasing up your legs or something like that, you probably wouldn't want to light one up again."* -**P109, 18–24, low motivation, man (Fig 4, sketch F)**.

*"You then forcibly put them [family] there, and they can't leave, and you can't help them. You then see a sped-up version of them decaying, while connected to constantly burning cigarettes. You can also have jump scares of their faces becoming completely distorted."* -**P125, 18–24, low motivation, woman (Fig 4, sketch E)**

Some participants felt that content which was too realistic would not be as impactful as something scary and dramatic.

*"I don't know how a virtual reality that's too similar to real life would change anything. It would have to be really shocking."* -**P125, 18–24, low motivation, woman.**

There were some concerns about whether it would be appropriate to show smokers such drastic content. The large majority of participants thought that this type of content could be acceptable if there were age restrictions (adults only), content warnings or mental health screenings beforehand (for example, excluding people with a pre-existing condition such as post-traumatic stress disorder).

**Theme 4- VR Content execution: Exploit VR's technological capabilities.** A handful of participants highlighted that the use of VR should not feel like a gimmick. Rather, the use of VR to promote quitting should make the most of VR's novel technology and feel genuinely different from traditional forms of messaging such as leaflets.

*"Yeah, I feel like a lot of VR studies or treatments take interventions that are already available, and they just translate them into VR, without necessarily making the most of what VR has to offer, but just using VR as this sexy, new intervention. . . What does VR add to the intervention that you wouldn't be able to get off other platforms?"* -**P87, 25–34, low motivation woman**.

*"I think it's important that it's tailored to the individual. If there was a way that you could enter what your personal reasons for quitting are or maybe the things that make you smoke when you're trying to quit- those kinds of things- because otherwise, I think it might just end up being a leaflet in VR form."* -**P103, 18–24, low motivation, man.**

After the VR demonstrations, many participants offered suggestions on how to increase its immersive quality. Suggestions included sound effects, being able to move around and interact with the virtual environment and prompts to let the user know it is a 360-degree environment. With the two previous content themes, the assumption is that the more immersive the VR experience is, the more pronounced the effects of the content and the more likely they would be to prompt a quit attempt. Additionally, some participants discussed the possibility of having

a VR scenario that could be tailored to match the users' motivation level and specific goals, for example, stopping smoking completely versus cutting down gradually.

*"Maybe if you could choose a range, like "I want to quit cold turkey" or "I want to quit in the next year. How can I get there?" versus just, you know, "just stop smoking and quit right now""* -**P136, 18–24, low motivation, woman.**

**Theme 5- Smoking cessation VR as a health service treatment.** All participants, except for one, enjoyed the VR demonstrations and most were amenable to the idea of using VR for smoking cessation. During the VR demonstration, there were no adverse events such as motion sickness. More than half of the participants (57.14%) had never used VR more than once before. Those with VR experience were mostly men. Only two participants owned a headset- participant 1 who mainly played video games with his headset and participant 108 whose sons used it for games too. Some of the older participants were more sceptical about using VR generally and for quitting smoking, which they expressly attributed to their age.

*"I wonder if it is something to do with the fact that because we're not digital natives, it doesn't come naturally to us in a way that it would for maybe a kid who's been using an iPad and things"* -**P165, 35–44, high motivation, woman**.

*"It [using the VR game] was a bit of a trial because I'm from a very older generation and family. So, I'm very new to this new way of doing things, not really into the technology world, you know"* -**P130, 35–44, low motivation, other.**

Given that all participants, except two, did not own a VR headset, there were concerns about the widescale availability of such an intervention. The main barriers identified for using VR at home were the cost of headsets and differential ease of use. During the VR demonstration, some participants found the headset and controllers easy to use, while others initially struggled.

*"Yeah, it was good fun and easy to use".* -**P17 35–44, low motivation, man**.

*"I don't know if it's just that I'm bad at gaming controls, that I found it a bit confusing at first. I mean, I think once you get the hang of it, it's really fun".* -**P136 18–24, low motivation, woman**.

*"I like it, even though. . . I didn't get how to use it straight away"* -**P92 18–24, low motivation, woman.**

Some participants suggested that a smoking VR scenario should first have a tutorial page to familiarise less experienced users with the controllers.

Community locations such as shopping centres, community centres, and libraries were suggested as potential locations for implementation. However, two participants in the last focus group (both women) were concerned about the safety of using VR in public locations. Since VR headsets block cues from the real world, this brought about a feeling of vulnerability.

*"Yes exactly! As a woman in the world, I am not taking away my ability to gauge the situation around me in a public place when everyone else is going to be able to."* -**P165, 35–44, high motivation, woman.**

Overall, participants preferred implementing the VR in primary care settings such as pharmacies, doctors' offices, and dental surgeries. Some participants previously had negative experiences with cessation services. This included a lack of available personnel and receiving advice that felt "basic", "patronising" and "condescending". They felt VR could be a useful tool to add on to existing services.

*"So yeah, put them [the VR] in the dentist and say, "Half the people you see are smokers" and get them to actually help their patients rather than them just saying that you've got to give up smoking and do this and do that"* -**P19, 45–54, high motivation, woman.**

In addition, some participants felt that placing the VR in a primary care setting would add to its legitimacy as a "treatment"- something that could be "prescribed", rather than an entertainment game. They wanted an intervention that was evidence-based and felt this would make sceptical smokers more willing to try it.

*"Also, if it's not, like, 100% scientifically approved, or it doesn't have a base. . . Like for people who grew up in the 90s, it could be a bit "funny". Like in "Friends" there was that episode where he would listen to that tape while he was sleeping so he could quit smoking. . . if it's not scientifically approved or very experimental, I think people would be sceptical"* -**P2, 35–44, high motivation, man**.

*"Yeah, I don't think it could be a stand-alone thing. It should go in conjunction with something else. Obviously, there are going to be people who a sceptical and it could give them a bit more faith in this potential VR-stop-smoking thing. If there's someone there talking them through that process and giving them advice, it might make them more willing to try it in the first place."* -**P163, 18–24, low motivation, other gender identity.**

**Theme 6**- **What happens after: Using VR for long-term cessation support.** Around half of the participants had previously tried to quit smoking. When discussing past and future quit attempts, participants mentioned environmental and social smoking cues that they commonly faced which would decrease the success of their quit attempt. Examples of these triggers included smoking in response to stress, anxiety, or seeing other people smoke in social settings such as pubs or festivals. A question posed by two participants was "What happens after the VR?". While the type of VR intervention proposed in this study may prompt a quit attempt, supporting the quit attempt falls outside its remit. This concerned participants.

*"In between these virtual reality places, where you're going to go and try it out, what happens in-between? You can't get one of them every half-hour."* -**P67, 64+, high motivation, man**.

*"I could come from that lovely community centre after going on VR, then I see [redacted- person's name] from up the road and it's a wrap. Or before I've reached the end of the road there's the smell of cigarettes coming through the air and then that's it".* -**P108, 35–44, low motivation, woman.**

Some participants proposed a long-term VR intervention that focuses on providing cessation advice and support. Some participants expressed a desire for more advice on the alternatives to smoking and advice on how to cut down (compared to stopping completely).

*"One thing, though, rather than say "Stop, smoking", "Finding an alternative" to stop smoking. Cos when they quit, they can't go back to cigarettes, so what's the alternative to smoking?*

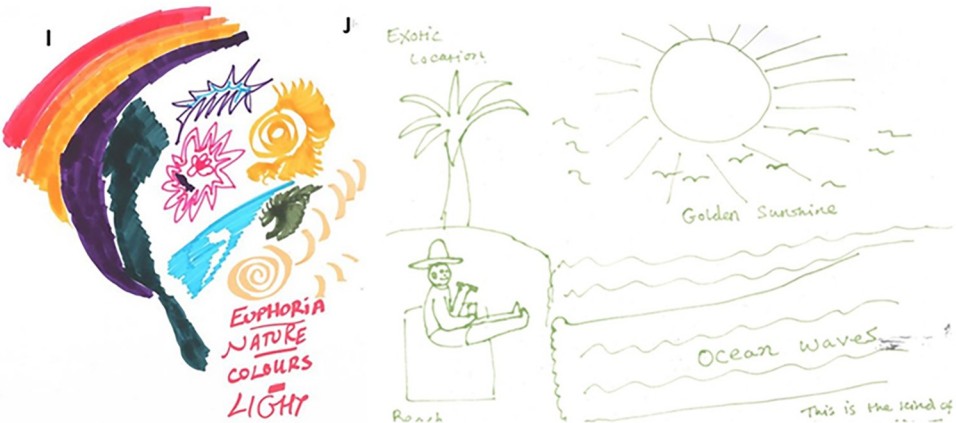

**Fig 5. Theme 6- Using VR for long term smoking cessation support, meditation image selection.**

*Let's say vaping or those. . . tobacco. . . what are the called. . . Nicorette products. . . So, if they're having that urge, let's say a relapse, they can just use an alternative, rather than go back to a cigarette." -***P3, 35–44, high motivation, man.**

Three participants highlighted that anxiety can be a trigger for smoking and suggested using VR to provide meditation or pleasurable images that could reduce anxiety and consequently reduce smoking levels.

*"VR, if it could help someone with anxiety or trouble sleeping, anything that resonates with the nervous system, could resonate with quitting smoking." -***P2, 35–44, high motivation, man.**

Fig 5 shows some of the content drawn by participants. In these sketches, there was an emphasis on "exotic locations", "golden sunshine", "colours" and "euphoria".

## Discussion

### Summary of findings

Using focus groups and art-based methods, this study aimed to explore smokers' views on using VR to encourage a quit attempt and identify what content could potentially be acceptable and relevant to this target group. We developed six themes concerning: the VR content suggested by participants, the rationale behind it, its technological execution, potential widescale implementation and potentially using VR for long-term quitting support. Overall, our results build upon previous qualitative work on the barriers and facilitators of quit attempts and smoking cessation media content.

In this study, most participants, especially the younger smokers, minimised the negative health effects of smoking and saw it as a problem for the distant future. In a previous qualitative study in the UK, smokers acknowledged the health risks of smoking but rationalised their continued smoking [52]. Similarly, in Tombor and colleagues' work creating typologies of smokers, those who had never made a quit attempt ("committed smokers") minimised the health consequences of smoking and downplayed their disease susceptibility [51]. A recent meta-ethnography of young people's pathways in and out of smoking revealed that younger smokers felt they were still young enough to not need to worry about cancer and saw quitting

smoking as a task for the future [61]. Our study corroborates these findings and extends them by suggesting content that could target these beliefs. This theme acted as the rationale for the two key themes on suggested VR content, on the basis that such content could accelerate a moment of awakening to the future health consequences.

The majority of the suggested VR content drew on the conventions of "coming of age" stories and involved looking into the future to see the benefits of smoking cessation (inspiring hope) or the negative outcomes of continued smoking (inspiring regret). There was no consensus among participants on whether positive or negative content was preferred. A recent randomized controlled trial evaluated a 90-minute documentary film that was developed in line with PRIME theory and focused on positive content framing [62]. The film included positive testimonials from ex-smokers, to boost motivation and promote a positive ex-smoker identity. The results indicated that the film did not increase the rate of quit attempts when compared to no intervention or a control film. However, an analysis of televised anti-tobacco media campaigns in England found that while both positive and negative messages increased the rate of calls to quitlines, positive campaigns were more effective [63]. As suggested by some participants, a VR scenario could potentially incorporate negative and positive outcomes and juxtapose them for added emphasis.

Family members were often drawn as supporting characters in the suggested VR scenarios. The COM-B model of behaviour highlights how motivation can change over time. Participants cited family as being a key influence in changing their motivation to quit. Similarly, in a qualitative study of smokers, ex-smokers and cessation professionals in the UK, family members and relationships were cited as one of the main factors in increasing motivation to quit [64]. In this present study, the influence of family either took the form of "desire" (wanting to quit because they valued more time with their family) or "obligation" (quitting because of guilt or because family members wanted them to). Some participants in the focus groups argued against quitting out of obligation and felt that quit attempts should be done for oneself rather than for family members (i.e., be intrinsically motivated). PRIME theory distinguishes between different facets of a person's motivational system and posits that obligation alone (quitting because you ought to) is insufficient to prompt a quit attempt without a genuine "want". In Tombor and colleagues' typology, "committed ex-smokers" had greater intrinsic motivation than "pragmatic ex-smokers"[51]. In a cross-sectional study of English smokers between 2008 to 2009, 39.0% of the sample believed they ought to quit smoking (obligation) compared to 29.3% who wanted to quit smoking (desire). However, only desire independently predicted quit attempts at 6 months [65]. A more recent study of ex-smokers combined both concepts into a single measure to assess desire over obligation in predicting continued abstinence. The results suggested that ex-smokers with greater desire over obligation had similar or lower levels of continued abstinence at 12 months follow-up, compared to people with lower desire over obligation [66]. Regarding, the implications for the development of a VR smoking cessation scenario, it does not follow that it must be an either/ or approach, focusing solely on desire or solely on obligation. Family members could be used as supporting characters to increase the emotional salience of the scenario, with both aspects of motivation targeted.

Additionally, some participants suggested content that placed a particular focus on graphic imagery, designed to elicit fear and anxiety. The drawings depicted horror-style images that showed the extreme effects of smoking and often went beyond the natural health consequences. In line with narrative transportation theory, the benefit of this type of content is that it could reduce the level of counter-arguing as these were not realistic events or stories based on rational arguments [20,21]. However, with fear appeals and graphic images, there is the potential for viewers to become defensive or inattentive to avoid experiencing negative emotions [67]. Evidence suggests that fear appeals are most effective when the target audience is

convinced of their ability to perform the behaviour [67]. This indicates that a VR scenario based on graphic images should be paired with content to boost self-efficacy. Also, while there is some debate on the ethicality of showing content designed to trigger fear or anxiety, the consensus is that fear appeals are appropriate when used to promote behaviours or interventions with a strong evidence base such as smoking cessation [67]. In this study, participants suggested age restrictions, content warnings and mental health screenings to avoid any unintended harm.

Regardless of the type of content suggested, there was a theme relating to its execution and how any VR intervention should genuinely feel different from other forms of messaging (for example a leaflet) and make the most of its unique technological capabilities. A potential downside of VR is its ability to induce motion sickness or cybersickness in users. This could lead to uncomfortable viewer experiences, especially if graphic content were to be used [68]. In this study, no participants spontaneously reported any adverse events during the VR demonstration. However, the views of experts in VR design will be sought to balance any potential harms with maximising engagement. Overall, participants enjoyed the demonstration and had favourable views towards using VR to encourage a quit attempt. Most participants with previous VR experience were men, possibly reflecting wider gender patterns of interest in VR. Previous UK survey data indicated that a higher proportion of men compared to women were interested in experiencing VR (55% vs 40%) [17].

Given the cost and low rates of headset ownership, participants thought that a smoking VR scenario would be most useful if available in healthcare settings such as dentists' and doctors' offices or pharmacies. Some suggest that it could be a beneficial adjunct to existing smoking cessation interventions (e.g., advice from a healthcare professional), which they had previously found unhelpful. While VR is still a relatively new technology, it is increasingly being incorporated into healthcare settings, for example, vaccination clinics, rehabilitation centres and mental health hospitals [69]. As part of this study's larger project, a future study with smoking cessation professionals (such as doctors, stop-smoking advisors, and pharmacists) will explore their opinions on the possible barriers and facilitators of incorporating VR into a healthcare setting, including accessibility issues and the need for specialised equipment.

In this study, participants were concerned about what would happen after using a VR scenario to motivate them to quit smoking. Even if the VR scenario was successful in prompting a quit attempt, participants were worried that the attempt would be unsuccessful if they encountered different smoking cues and stressors afterwards. They saw the VR intervention as a temporary fix. The transtheoretical model of smoking cessation highlights the importance of behavioural support and NRT during the maintenance stage to prevent relapse [8]. Cross-sectional data from 2007–2018 suggests that on average only 16.2% of quit attempts in smokers in England are successful [70]. Other studies have used VR to deliver long-term interventions such as cue exposure therapy and mindfulness therapy [71,72]. However, the effects of VR cue exposure therapy on long-term abstinence have been inconsistent across trials [73].

Some participants suggested using VR for long-term cessation support. The content drawn involved pleasurable images that were "exotic," "euphoric" and brightly coloured. The aim was to reduce anxiety and stress, which were common triggers for smoking. However, this type of content may also bring its own set of ethical concerns. Numerous scholars and philosophers are concerned with the psychological and existential dimensions of VR [74]. Some authors argue that prolonged exposure to immersive technologies and virtual worlds could lead to users feeling dissatisfied with the real world and potentially developing depressive symptoms [75]. This would be counter-productive to the intentions of the participants who suggested this type of content. A study conducted in 2009 found that exposure to VR increased dissociative experiences in participants and lessened their sense of presence in actual reality [75].

However, a description of the VR content shown was not included in that study. While outside the remit of this project, future work could explore the usefulness and implications of content based on pleasurable images in treating substance use, (including and beyond smoking) and mental health more generally.

Furthermore, given the low ownership of VR headsets [18], an intervention that involves repeated VR sessions would potentially be more expensive, more time-consuming, and less scalable. An alternative approach could be to combine a VR intervention aimed at encouraging quit attempts with low-cost, follow-up, digital support such as a smartphone app [76]. For example, an ongoing US trial of a VR intervention aimed at motivating quit attempts follows up the VR scenario, shown once to participants, with a four-week smoking cessation text messaging program [77]. For this study's wider project, future work could explore combining the eventual VR scenario with an evidence-based digital intervention such as a smartphone application or text messaging, to provide long-term cessation support.

## Strengths and limitations

This study has a strong theoretical underpinning, with multiple and innovative methods of data collection (focus groups and sketches) used. One of the main strengths of this study is the varied sample. Although it skewed younger, it was mostly balanced according to gender, occupational status, VR experience, and previous use of a digital smoking cessation aid. It was also ethnically diverse. While there was a slightly higher proportion of participants with a low motivation to quit, this mirrors current patterns in England, where the majority of smokers are not highly motivated to quit [2]. However, the results of this study may not be generalisable to very motivated smokers. While we recruited fewer participants than originally planned, when analysing the last focus group, no new themes were developed. However, a larger sample size would have ensured true theoretical saturation.

We were unable to reach the goal of having half the sample from a less advantaged SEP- while occupations were varied, the majority of participants had higher educational attainment (post-16 qualifications). The sample was skewed towards younger, digitally connected participants. Also, as the study was predominantly advertised through online methods, there is a possibility that this study does not include the viewpoints of those who are most digitally excluded and without regular internet access. This may impact the generalisability of findings to all subgroups, especially those who are older or most socially disadvantaged. Additionally, we were unable to segment the focus groups by motivation to quit. Some authors suggest that more homogenous focus groups increase participants' ability to speak freely [30]. This would increase the credibility of the data produced (i.e., the extent to which findings represented the true ideas of the participants) [57]. As smoking can often be stigmatised due to its negative health effects, we initially posited that smokers with lower motivation to quit may feel inhibited if in a room with high-motivation smokers expressing anti-tobacco views [78]. For example, in this study, only one participant expressed an explicitly positive smoker identity. It is possible that other participants felt unable to do so. However, the majority of participants were open about their struggles to quit or their plans to continue smoking, which suggests that there was some level of comfort.

Due to a combination of oversubscriptions and last-minute cancellations, rather than 6 participants per focus group, the sizes ranged from 2 to 10. We found that the durations of the focus groups were similar (approximately 1.5 to 2 hours), regardless of size, indicating that the smaller groups had sufficient interaction. However, for the larger group (n = 10), there may not have been enough time for each member to adequately participate and in equal proportion.

## Implications for future research and practice

**Design requirements.**  Based on the results of this study we suggest the following:

- VR content emphasising the future health consequences of smoking or the benefits of cessation.

- The use of graphic imagery to evoke emotion when highlighting the negative impact of smoking.

- Emotive storytelling through techniques such as family members as supporting characters in the narrative.

- Options that allow the user to interact with and be immersed in the virtual environment such as prompts to make you move around in the virtual environment and sound effects.

- Content that can be tailored to meet the needs of the specific user.

- Follow-up cessation support after the VR content is viewed.

Nevertheless, co-design workshops with future intervention users rarely result in service-ready solutions [79]. As part of the larger project, future work will take these VR design requirements and refine them in a second study with intersectoral experts in smoking cessation and VR design. We will take these ideas and explore which specific behaviour-change techniques and technological features can feasibly be incorporated into them. Additionally, while one large-scale trial is in progress in the USA, future work will be needed to assess the long-term efficacy of VR interventions aimed at increasing motivation to quit in smokers [77]. Although this study was focused specifically on VR, the content suggested could also be used to inform new smoking cessation campaigns using mediums such as television and social media videos.

## Conclusion

In conclusion, participants in this study had favourable views towards using VR to encourage smoking quit attempts. Suggested VR content either focused on the benefits of cessation or the negative consequences of not quitting. Participants thought that a VR intervention would be best placed in a healthcare setting but highlighted the need for follow-up support. Future research will refine the ideas suggested in this study with input from professional stakeholders in VR, digital health, and smoking cessation.

## Supporting information

**S1 Table. REDCap screening survey.**
(DOCX)

**S2 Table. Question route for focus groups.**
(DOCX)

**S1 Fig. Example of a coded drawing.**
(DOCX)

## Acknowledgments

Tugba Cetin (MSc student) supported data collection for this study.

## Author Contributions

**Conceptualization:** Tosan Okpako, Dimitra Kale, Olga Perski, Jamie Brown.

**Data curation:** Tosan Okpako.

**Formal analysis:** Tosan Okpako.

**Methodology:** Tosan Okpako.

**Project administration:** Dimitra Kale, Olga Perski, Jamie Brown.

**Supervision:** Dimitra Kale, Olga Perski, Jamie Brown.

**Writing – original draft:** Tosan Okpako.

**Writing – review & editing:** Tosan Okpako, Dimitra Kale, Olga Perski, Jamie Brown.

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
