## [Decision Letter · Decision Letter 0]

4 Jan 2024

PDIG-D-23-00484

Developing content for a virtual reality scenario that motivates quit attempts in adult smokers: A focus group study with art-based methods.

PLOS Digital Health

Dear Dr. Okpako,

Thank you for submitting your manuscript to PLOS Digital Health. After careful consideration, we feel that it has merit but does not fully meet PLOS Digital Health's publication criteria as it currently stands. Therefore, we invite you to submit a revised version of the manuscript that addresses the points raised during the review process.

Please submit your revised manuscript within 60 days Mar 04 2024 11:59PM. If you will need more time than this to complete your revisions, please reply to this message or contact the journal office at digitalhealth@plos.org. Please include the following items when submitting your revised manuscript:

We look forward to receiving your revised manuscript.

Kind regards,

Haleh Ayatollahi

Section Editor

PLOS Digital Health

Journal Requirements:

Additional Editor Comments (if provided):

Reviewers' comments:

Reviewer's Responses to Questions

**Comments to the Author**

1. Does this manuscript meet PLOS Digital Health’s publication criteria? Is the manuscript technically sound, and do the data support the conclusions? The manuscript must describe methodologically and ethically rigorous research with conclusions that are appropriately drawn based on the data presented.

Reviewer #1: Yes

Reviewer #2: Yes

2. Has the statistical analysis been performed appropriately and rigorously?

Reviewer #1: N/A

Reviewer #2: No

3. Have the authors made all data underlying the findings in their manuscript fully available (please refer to the Data Availability Statement at the start of the manuscript PDF file)?

Reviewer #1: Yes

Reviewer #2: Yes

4. Is the manuscript presented in an intelligible fashion and written in standard English?

Reviewer #1: Yes

Reviewer #2: Yes

5. Review Comments to the Author

Reviewer #1: The paper presents a valuable exploration of using virtual reality (VR) for smoking cessation interventions, incorporating focus groups and art-based methods to inform content development. The paper is well-structured, and the abstract effectively communicates the study's objectives, methods, and key findings. In my opinion, the title is quite long, I think it might be interesting to shorten it a bit.

The use of focus groups and art-based methods adds an innovative dimension to the study, offering a unique approach to gather insights from the target population. Also, the authors have gone to great lengths to show all the steps of the method in detail. 

In any case, I wonder if instead of presenting the characteristics of the sample in general in Table 1, it wouldn't be worth to have a detailed table with information on each participant, not least because the authors explain that they wanted to segment the focus groups by level of motivation, but then they can't do it. I don't see this as a critical point but rather, for example, the imbalance in the number of participants in the focus groups (e.g. group 2 had 10 participants and group 4 had 2 participants). In addition, although the paper mentions the inclusion of participants from ethnic minorities and women, it would be beneficial to elaborate on how these demographic factors might have influenced the findings. Are there any notable patterns or differences in perspectives based on ethnicity or gender?

I missed having, in a very concrete and pragmatic way, some guidelines and requirements for the development of this VR intervention based on these findings.

Reviewer #2: ****Overall assessment: The strengths of the study lie in its diverse sample, theoretical grounding, and exploration of ethical considerations. However, limitations related to sample diversity, potential biases, and concerns about long-term efficacy should be considered in interpreting the findings. The issues related to socioeconomic disparity, potential digital exclusion bias, and the homogeneity in focus groups should be carefully addressed. Additionally, providing more clarity on certain aspects, such as the potential long-term efficacy of VR interventions, could strengthen your manuscript.

Consider revising the manuscript to address these weaknesses, providing more robust explanations, and possibly conducting additional statistical analyses and providing more context where necessary. This will help ensure the credibility and generalizability of your findings.

Overall Strengths:

Innovative Approach: The use of virtual reality (VR) for smoking cessation is an innovative approach, suggesting a willingness to explore novel methods for health interventions.

Theoretical Framework: The study is grounded in health behavior theory, narrative transportation theory, and the Person-Based Approach, showcasing a strong theoretical foundation for the research.

Diversity in Sample: The inclusion of participants from an ethnic minority (42.8%) and a balance between genders (42.9% women) enhances the generalizability of the findings, making them more applicable to a diverse population.

Mixed Methods Design: Combining focus groups with art-based methods (participant sketches) provides a comprehensive understanding and allows for a multi-faceted analysis of the data.

Thorough Analysis: The use of reflexive thematic analysis demonstrates a robust analytical method, ensuring a systematic and comprehensive exploration of the collected data.

Participant Involvement: The involvement of participants in generating themes and ideas for VR content adds depth and relevance to the intervention, making it more likely to be accepted by the target population.

Consideration of Implementation Settings: The consideration of healthcare locations, such as doctors' offices, for VR implementation addresses practical concerns like low home ownership of headsets, indicating a thoughtful approach to real-world application.

Overall Weaknesses:

Limited Sample Size: The study involved only four in-person focus groups with a total of 21 participants. A larger sample size might be beneficial for a more robust understanding of diverse perspectives and potential themes.

Demographic Representation: While there is diversity in gender and ethnicity, other demographic factors such as socioeconomic status and educational background are not explicitly mentioned. These factors could impact the generalizability of the findings.

Focus on Fear Appeals: While fear appeals were suggested as a component, relying too heavily on anxiety-inducing content may raise ethical concerns and could potentially lead to unintended negative consequences.

Assumption about VR Legitimacy: Assuming that implementing VR in healthcare locations would make it appear more legitimate without direct evidence might be a premature conclusion. Further validation of this assumption is necessary.

Need for Expert Input: While participants generated ideas, the study acknowledges the need for future research to refine these ideas with experts in VR design and smoking cessation. This raises questions about the feasibility and effectiveness of the proposed VR content.

Limited Timeframe: The relatively short timeframe of the focus groups (between July and August 2023) might not capture potential changes in attitudes or perceptions over a more extended period.

Motivation to Quit: The study mentions that more than half of the participants had low motivation to quit (61.0%). This might impact the generalizability of findings to individuals with higher motivation to quit smoking.

**Introduction: 

Strengths:

Clear Problem Statement: The introduction effectively highlights the serious health risks associated with smoking and the existing challenges in motivating smokers to quit. This provides a clear context for the study.

Innovative Use of VR: The integration of virtual reality (VR) technology for smoking cessation is presented as a novel and promising approach, demonstrating a forward-thinking perspective.

Evidence from Pilot Studies: Reference to previous pilot studies testing VR for encouraging quit attempts adds credibility to the potential effectiveness of the proposed intervention.

Theoretical Framework: The incorporation of the COM-B model and PRIME theory provides a strong theoretical foundation, helping to understand the underlying factors influencing smoking behavior.

Consideration of Emotional Response: The emphasis on the need for emotional responses in prompting quit attempts aligns with psychological theories, adding depth to the study's rationale.

Connection to Healthcare Trends: The acknowledgment of VR's emerging impact in healthcare and its predicted influence on the NHS workforce by 2035 strengthens the argument for the relevance of VR in smoking cessation interventions.

Acknowledgment of Digital Inequalities: The discussion about the potential risk of digital inequalities and the importance of ensuring VR acceptability across socioeconomically disadvantaged groups demonstrates a consideration for equitable healthcare solutions.

Integration of Narrative Transportation Theory: The incorporation of narrative transportation theory to enhance emotional responses through storytelling adds a nuanced layer to the proposed VR scenario.

Person-Based Approach (PBA): The use of the PBA aligns with current best practices in public health research, emphasizing formative development work and grounding the study in the perspectives of the target population.

Co-Design and Participant Involvement: The emphasis on co-design and participant involvement as "experts of their experience" reflects a participatory approach, increasing the relevance and potential acceptability of the intervention.

Weaknesses:

Limited Justification for VR Choice: While the benefits of VR are highlighted, there is a lack of explicit justification for choosing VR over other digital health interventions. A brief comparison could strengthen the rationale.

Assumption of Greater Emotional Response in VR: The claim that VR can produce stronger cognitive and emotional responses than traditional media lacks direct evidence within the context of smoking cessation interventions.

Digital Inequalities Generalization: While the study acknowledges digital inequalities, it generalizes the risk without specific evidence or data related to the target population's access to VR technology.

Complex Terminology: The introduction contains complex terminology related to behavior models (COM-B, PRIME) and theories, potentially making it less accessible to readers without a background in these concepts.

Lack of Preliminary Findings: The introduction doesn't provide any preliminary findings from the pilot studies or initial data collection, leaving the reader curious about the initial insights that prompted the current research.

Minimal Discussion on Ethical Considerations: There is limited discussion about potential ethical considerations associated with using VR, especially when incorporating fear appeals or anxiety-inducing content.

Potential Bias in Participant-Generated Ideas: While participant involvement is a strength, there may be a potential bias in the ideas generated, and it's crucial to acknowledge this and discuss potential implications.

Wordiness: Some sentences are lengthy and may benefit from simplification for improved readability.

In summary, the introduction effectively communicates the problem, the innovative use of VR, and the theoretical foundations. However, some weaknesses, such as the limited justification for VR, complex terminology, and the lack of preliminary findings, should be addressed to enhance the overall clarity and impact of the study.

**Research Questions: 

Strengths:

Clearly Stated Research Questions: The research questions are explicitly stated, providing a clear focus for the study. This clarity enhances the reader's understanding of the study's objectives.

Alignment with Study Objectives: The research questions align well with the study's overall aim of developing VR content that is acceptable to smokers and addresses their evaluative beliefs about smoking and quitting.

Specificity: The questions are specific and directly related to the intended outcomes of the study, facilitating a straightforward approach to data collection and analysis.

Relevance to Target Population: The research questions directly involve the target population (adult smokers), ensuring that the study outcomes are applicable and meaningful to the individuals for whom the intervention is intended.

Combination of Qualitative and Acceptability Aspects: The questions encompass both qualitative aspects (e.g., beliefs and views) and the concept of acceptability, providing a holistic approach to understanding participants' perspectives.

Weaknesses:

Limited Scope: The research questions are relatively narrow and might benefit from expansion to include a broader range of factors influencing smoking behavior or perceptions of VR interventions.

Potential Bias in Framing: The framing of the questions assumes a positive stance toward VR content, potentially introducing bias. A more neutral or exploratory framing could encourage a wider range of responses.

Assumption of Homogeneity: The questions imply a degree of homogeneity within the adult smoker population, assuming that thei

---

## [Decision Letter · Decision Letter 1]

12 Mar 2024

PDIG-D-23-00484R1

Developing content for a virtual reality scenario that motivates quit attempts in adult smokers: A focus group study with art-based methods.

PLOS Digital Health

Dear Dr. Okpako,

Thank you for submitting your manuscript to PLOS Digital Health. After careful consideration, we feel that it has merit but does not fully meet PLOS Digital Health's publication criteria as it currently stands. Therefore, we invite you to submit a revised version of the manuscript that addresses the points raised during the review process.

Please submit your revised manuscript within 60 days May 11 2024 11:59PM. If you will need more time than this to complete your revisions, please reply to this message or contact the journal office at digitalhealth@plos.org. Please include the following items when submitting your revised manuscript:

We look forward to receiving your revised manuscript.

Kind regards,

Haleh Ayatollahi

Section Editor

PLOS Digital Health

Journal Requirements:

Additional Editor Comments (if provided):

Reviewers' comments:

Reviewer's Responses to Questions

**Comments to the Author**

1. If the authors have adequately addressed your comments raised in a previous round of review and you feel that this manuscript is now acceptable for publication, you may indicate that here to bypass the “Comments to the Author” section, enter your conflict of interest statement in the “Confidential to Editor” section, and submit your "Accept" recommendation.

Reviewer #1: All comments have been addressed

Reviewer #3: All comments have been addressed

Reviewer #4: (No Response)

Reviewer #5: (No Response)

Reviewer #6: All comments have been addressed

2. Does this manuscript meet PLOS Digital Health’s publication criteria? Is the manuscript technically sound, and do the data support the conclusions? The manuscript must describe methodologically and ethically rigorous research with conclusions that are appropriately drawn based on the data presented.

Reviewer #1: Yes

Reviewer #3: Yes

Reviewer #4: Partly

Reviewer #5: No

Reviewer #6: Yes

3. Has the statistical analysis been performed appropriately and rigorously?

Reviewer #1: Yes

Reviewer #3: N/A

Reviewer #4: N/A

Reviewer #5: No

Reviewer #6: Yes

4. Have the authors made all data underlying the findings in their manuscript fully available (please refer to the Data Availability Statement at the start of the manuscript PDF file)?

Reviewer #1: Yes

Reviewer #3: Yes

Reviewer #4: No

Reviewer #5: Yes

Reviewer #6: Yes

5. Is the manuscript presented in an intelligible fashion and written in standard English?

Reviewer #1: Yes

Reviewer #3: Yes

Reviewer #4: No

Reviewer #5: Yes

Reviewer #6: Yes

6. Review Comments to the Author

Reviewer #1: (No Response)

Reviewer #3: The recommended revisions have been addressed satisfactorily.

Reviewer #4: There is no rationale for the chosen sample size.

There is a skew toward younger, digitally connected individuals.

Discuss how the researcher's factors were managed during the focus groups and the reflexivity strategies employed during data analysis.

Elaborate on potential limitations arising from pre-registered protocol modifications and how these changes affect the study's validity and generalizability.

Describe the recruitment process and provide information on focus group moderation techniques and strategies to mitigate bias.

Discuss limitations related to the use of VR technology, such as accessibility issues and the need for specialized equipment.

Discuss specific measures implemented to minimize harm. Is there any mental health support for participants experiencing distress?

Discuss safeguards to protect vulnerable participants and ensure equitable access to VR interventions.

Ethical oversight should extend beyond the study period to address potential long-term effects on participants' well-being and autonomy. Were there follow-up assessments and ethical monitoring post-intervention?

Discuss potential synergies between VR interventions and pharmacotherapy.

Discuss the safety concerns of integrating VR interventions into smoking cessation treatment plans and how it would help pharmacists.

Discuss potential counterarguments and alternative perspectives on the ethical and existential dimensions of VR-based smoking cessation interventions.

Reviewer #5: Authors have proposed an interesting topic, but the methods chosen to treat it are neither rigorous nor appropriate.

The results obtained are not significant, as they derive from the comparison of non-homogeneous groups in terms of number, age, daily cigarette consumption and familiarity with VR environments, all elements that characterise and heavily influence the results.

Authors are encouraged to extend their research and show robust and rigorous results.

Furthermore, the different desire to quit smoking in the participating subjects, the support received from the family context, and the custom motivational elements present in the VR environment are all factors that influence in a different, but not independent way, the level of patients' engagement with respect to the chosen method. This prevents the motivational results from being delegated solely to the VR environment chosen for each patient. Authors are invited to standardise the chosen approach, so that there can be no doubt about the causes that determine the results shown.

Reviewer #6: The significance of health promotion in the field of smoking addiction is a topic of growing interest, particularly in light of new technologies such as virtual reality (VR). Smoking addiction poses one of the major challenges to public health globally, with significant implications for morbidity and mortality related to cardiovascular, respiratory, and oncological diseases. In this context, the adoption of innovative digital health tools, such as VR, opens new frontiers for health promotion and the treatment of smoking addiction.

VR, in particular, has shown to offer unique opportunities for intervention in smoking cessation, thanks to its ability to simulate controlled environments where smokers can be exposed to situations that typically evoke the desire to smoke (cue exposure therapy), but in a safe context where they can practice coping strategies without the risk of relapse.

Your work is very interesting because it has a bottom-up user-centered focus and deserves an equally interesting theoretical explanation. For this reason, I would include in the initial part references to past attempts already made in the field of smoking cessation with digital health tools, also referring to the most used psychological theories in the field of addiction treatment, and also giving the abstract a dry and clear writing that the rest of the manuscript also has.

R94-110: The works of Michie et al. are very interesting in the field of theories that explain change, addiction, and behavior, but it could also be interesting for the reader and strengthen the manuscript to know the theoretical references prior to the team's work, and I refer in particular to: the biopsychosocial model and theories of Health Psychology, Theory of Planned Behavior, Social Cognition, Transtheoretical Model, Social Reinforcement Model, and Self-Efficacy Model. These models are just an example that could make the introduction more complete and organic compared to the excellent work done later.

R251-252: A non-expert reader might find it interesting to read an explanation of the environments on which these VR games are based or a small note on the sources that report basic information released by the manufacturing company.

R363-364: At this point, it could be interesting to make a reference to the transtheoretical model to explain the importance of maintenance and avoidance of relapse.

R419: It does not seem that the link clearly and easily contains the package of sketches of the participants.

R519-520: An expert reader might find it interesting to know how many minutes the demonstration lasted. Generally, an inexperienced gamer, on the first use, has difficulty using the headset for more than 20/40 minutes consecutively. Was the demonstration in this range or was it lower or higher?

In a general reading, a less experienced reader might be interested in understanding the specific characteristics of the two headsets used, what components they are made of, and why those types of headsets were chosen, as well as the price range. Moreover, for the sample that had no experience with VR headsets, it would be interesting to understand whether the environments they were subjected to were realistic or not.

7. PLOS authors have the option to publish the peer review history of their article (what does this mean?). If published, this will include your full peer review and any attached files.

**Do you want your identity to be public for this peer review?** For information about this choice, including consent withdrawal, please see our Privacy Policy. 

Reviewer #1: None

Reviewer #3: Yes: Dr Sarah Markham

Reviewer #4: No

Reviewer #5: No

Reviewer #6: Yes: Andrea Moi

---

## [Decision Letter · Decision Letter 2]

16 Apr 2024

Developing content for a virtual reality scenario that motivates quit attempts in adult smokers: A focus group study with art-based methods.

PDIG-D-23-00484R2

Dear Miss Okpako,

We are pleased to inform you that your manuscript 'Developing content for a virtual reality scenario that motivates quit attempts in adult smokers: A focus group study with art-based methods.' has been provisionally accepted for publication in PLOS Digital Health.

Best regards,

Haleh Ayatollahi

Section Editor

PLOS Digital Health

Reviewer Comments (if any, and for reference):

Reviewer's Responses to Questions

**Comments to the Author**

1. If the authors have adequately addressed your comments raised in a previous round of review and you feel that this manuscript is now acceptable for publication, you may indicate that here to bypass the “Comments to the Author” section, enter your conflict of interest statement in the “Confidential to Editor” section, and submit your "Accept" recommendation.

Reviewer #1: All comments have been addressed

Reviewer #3: All comments have been addressed

Reviewer #4: All comments have been addressed

Reviewer #5: All comments have been addressed

Reviewer #6: All comments have been addressed

2. Does this manuscript meet PLOS Digital Health’s publication criteria? Is the manuscript technically sound, and do the data support the conclusions? The manuscript must describe methodologically and ethically rigorous research with conclusions that are appropriately drawn based on the data presented.

Reviewer #1: Yes

Reviewer #3: Yes

Reviewer #4: Yes

Reviewer #5: Yes

Reviewer #6: Yes

3. Has the statistical analysis been performed appropriately and rigorously?

Reviewer #1: Yes

Reviewer #3: N/A

Reviewer #4: N/A

Reviewer #5: N/A

Reviewer #6: Yes

4. Have the authors made all data underlying the findings in their manuscript fully available (please refer to the Data Availability Statement at the start of the manuscript PDF file)?

Reviewer #1: Yes

Reviewer #3: Yes

Reviewer #4: Yes

Reviewer #5: Yes

Reviewer #6: Yes

5. Is the manuscript presented in an intelligible fashion and written in standard English?

Reviewer #1: Yes

Reviewer #3: Yes

Reviewer #4: Yes

Reviewer #5: Yes

Reviewer #6: Yes

6. Review Comments to the Author

Reviewer #1: (No Response)

Reviewer #3: I am happy with the revisions made.

Reviewer #4: (No Response)

Reviewer #5: Authors have adequately addressed my comments.

Reviewer #6: I found your review and new comments very interesting. The comments of previous reviewers have already pointed out some of the limitations of this good work, but I still find it worth publishing for the interesting application of the model.

7. PLOS authors have the option to publish the peer review history of their article (what does this mean?). If published, this will include your full peer review and any attached files.

**Do you want your identity to be public for this peer review?** For information about this choice, including consent withdrawal, please see our Privacy Policy.

Reviewer #1: No

Reviewer #3: No

Reviewer #4: **Yes: **Hisham E. Hasan

Reviewer #5: No

Reviewer #6: **Yes: **Andrea Moi
